# Realistic Physiological Options to Increase Grain Legume Yield under Drought

**DOI:** 10.3390/plants12173137

**Published:** 2023-08-31

**Authors:** Thomas R. Sinclair, Michel E. Ghanem

**Affiliations:** 1Crop and Soil Sciences Department, North Carolina State University, Raleigh, NC 27695-7620, USA; 2UMR AGAP Institut, Université de Montpellier, CIRAD, INRAE, Institut Agro, F-34398 Montpellier, France; michel_edmond.ghanem@cirad.fr; 3CIRAD, UMR AGAP Institut, F-34398 Montpellier, France

**Keywords:** grain legumes, growth, limited transpiration trait, nitrogen fixation, transpiration, vapor pressure deficit

## Abstract

Increasing yield resiliency under water deficits remains a high priority for crop improvement. In considering the yield benefit of a plant trait modification, two facts are often overlooked: (1) the total amount of water available to a crop through a growing season ultimately constrains growth and yield cannot exceed what is possible with the limited amount of available water, and (2) soil water content always changes over time, so plant response needs to be considered within a temporally dynamic context of day-to-day variation in soil water status. Many previous evaluations of drought traits have implicitly considered water deficit from a “static” perspective, but while the static approach of stable water deficit treatments is experimentally congruous, the results are not realistic representations of real-world drought conditions, where soil water levels are always changing. No trait always results in a positive response under all drought scenarios. In this paper, we suggest two key traits for improving grain legume yield under water deficit conditions: (1) partial stomata closure at elevated atmospheric vapor pressure deficit that results in soil water conservation, and (2) lessening of the high sensitivity of nitrogen fixation activity to soil drying.

## 1. Introduction

Since humans sowed the first crop seeds, the hope of farmers has been that there will be sufficient water to obtain a grain yield. While management regimes, including irrigation, have resulted in progress toward this hope [1], until very recently, there has been little evidence of plant modifications for altering plant genetics in order to improve crop yield under drought [2]. In fact, during the last century, the push for ever increasing yields is likely to have resulted in greater water requirements and, consequently, an increased vulnerability to drought.

The development of technology to identify and transform genes related to plant response to water deficit, including grain legumes, has resulted in renewed optimism regarding the age-old hope of plant modification that can decrease crop vulnerability to drought. For example, a recent review by Gupta et al. [3] lists gene-linked physiological characteristics that have been hypothesized to result in such crop improvement. The challenge here, though, is that the fundamental physiology and especially the physics that make crop yields quantitatively dependent on the amount of water available to a crop have not changed. While plant modifications for static short-term water deficit may show advantages in controlled environments, the realities of grain legume production under temporally dynamic conditions in the field can severely limit, or even negate, the possible yield benefits of such trait alterations.

## 2. Qualitative Seminal Observations on Linkage between Yield and Water

Qualitative observations on the linkage between plant growth and transpiration can be traced back to 1699 when Woodward [4] published one of the first papers on plant physiology. A century ago, vigorous research was conducted worldwide, and a consistent conclusion was reached that plant growth and water loss were very closely related. One of the more prolific sets of experiments was reported in 1913 by Briggs and Shantz [5]. 

In 1958, de Wit [6] published an analysis of the data from many of the earlier studies, and he graphed plant growth vs. transpiration divided by an index of atmospheric humidity. As shown in Figure 1, for each crop species, de Wit found that there was a unique, linear correlation between these variables, including a slope that was insensitive to cultivar, environment, soil texture, soil fertility, and, importantly for this review, water regime. However, these empirical results offered no indications for possible alterations in the relationship between plant growth and water loss within a species.

## 3. Phenomenological Model of Passioura

In spite of de Wit’s analysis showing a close relationship between crop growth and water loss, there has been huge amounts of research over the last 50 years attempting to discover plant modifications that can obtain greater crop yield for the same amount of water input, and this is sometimes referred to by the catchy phrase, “more crop per drop”. (As an aside, this phrase is even trademarked by Lotus Foods Inc., Richmond, CA, USA (https://www.lotusfoods.com/more-crop-per-drop/, accessed on 27 August 2023) The enthusiasm for the “more crop per drop” idea seems to have been encouraged by a somewhat simple perspective of water use by crops. In the phenomenological model presented by Passioura [7], yield (Y) is presented as a function of water for transpiration (T), harvest index (HI), and plant water use efficiency (WUE):Y = T • HI • WUE(1)

This equation implies that simply increasing plant WUE will result in increased crop yield. 

The dependence of crop growth on transpiration is a major feature of the phenomenological model. This phenomenological expression, however, does not define some of the very basic physiological factors that influence crop gas exchange. In particular, the dominating influence of atmospheric humidity on the relationship, as identified by de Wit [6], is not even an explicit part of the phenomenological equation. As discussed in the following section, options to both fully understand and improve crop yield with limited water require a more mechanistic expression of plant water use based on crop gas exchange.

## 4. A Mechanistic Description of Water Use

Tanner and Sinclair [8] derived a fundamental relationship between plant growth and transpiration based on leaf gas exchange properties. Their derivation started with instantaneous expressions of leaf water vapor and CO_2_ exchange, and they were combined and integrated in order to describe crop yield (Y) as a function of transpiration (T). The result of the derivation was the following equation, which appears to be fairly simple:ʃ Y dt = HI • k • ʃ (T/VPD) dt (2)

The variables in Equation (2) are as follows: t is time, HI is harvest index, k is mechanistic coefficient expressed in pascal units (Pa) and explicitly defined by crop physical and physiological characteristics, and VPD is vapor pressure deficit (Pa). VPD is the water vapor pressure difference between the interior of leaves (which is very near saturation at leaf temperature) and the atmosphere surrounding the leaves. Tanner and Sinclair [8] presented the results of their derivation as an integral equation to reflect the temporal variability of both T and VPD. 

The structure of Equation (2) is fully consistent with that of de Wit’s analysis, except that the empirical slope found by de Wit is mechanistically defined in Equation (2) by the k variable. The analysis of de Wit indicates that k is constant within a species. Does the mechanistic definition of k found by Tanner and Sinclair [8] support this conclusion? The variability in the value of k is dependent on two key variables: (1) the energy content of the plant products, and (2) the CO_2_ concentration ratio between the leaf interior and the atmosphere surrounding the leaf. For grain crops, biochemical seed composition is generally narrowly restricted by the commercial uses for which the grain is grown. Consequently, changes in seed composition are not generally considered to be viable options for most crop species, including grain legumes, that will allow major increases in k. 

The CO_2_ concentration ratio in the definition of k is determined to a large extent by the C-assimilation pathway, i.e., C3 vs. C4. Only in cases where CO_2_ concentration of the leaf interior is unusually high relative to the expected ratios for C3 and C4 species will there be an opportunity to increase k. Since genotypes with high CO_2_ ratios would almost necessarily have a low growth rate and, therefore, low yields, a focus on lowering this ratio of such genotypes is likely an indication of a need to increase leaf gas exchange rate. Overall, it is anticipated that for existing high-yielding, commercial crop genotypes, the value of k for each species is limited to a narrow range that is consistent with the linear correlations that are found in the empirical analysis of de Wit [6]. 

## 5. Challenges of an Intuitive Perspective 

In recent times, the insights of the yield limitation imposed by a lack of water, as described above, have often been bypassed in favor of more intuitive perspectives on hypothesizing solutions to water deficit. That is to say, by modification of specific plant genes or physiological processes, it is assumed that crop yield under drought might be increased. Often these hypothesized approaches are based on static plant responses observed under stable environmental situations. In this section, options usually originating from an intuitive view of improving plant response to drought are discussed. An important framework of this discussion is the impact of such intuitive options on season-long, temporal dynamics of water use in the field based on a limited amount of water that is available for the entire growing season.

### 5.1. Crop Survival

There is little practical benefit for farmers to have annual grain-crop plants that have an increased ability to survive drought. Weather conditions that lead to severe drought and that threaten plant survival are a result of sustained periods of little or no rainfall. During such sustained severe drought, there is a continuous decreasing crop growth rate, which ultimately reaches zero growth as the crop enters the period when crop survival may be threatened. If the crop survives a prolonged severe drought, the overall growth of the crop will have been so severely decreased over many weeks of water deficit that the grain yield of the crop will be extremely low. Even if the ability of a crop to survive this type of life-threatening drought is increased, the resulting low yield is more than likely to be an economic disaster for the farmer. Our conclusion is that there is essentially no commercial market for drought-surviving crop genotypes, including grain legumes.

### 5.2. Osmotic Accumulation

Genotypes within crop species have been identified that either constitutively accumulate osmotic compounds or have increased accumulation as water deficits develop [9]. It has been asserted that such osmotic accumulation increases the hydrostatic pressure gradient in order to facilitate soil water uptake and/or allow beneficial maintenance of cell turgor as water deficits develop. However, the advantages of these hypotheses in the temporal dynamics of water use for crop production are not as straightforward as they are often presented. 

While the hydrostatic pressure gradient may be increased and may initially allow more rapid water uptake, the long-term advantage is not at all clear. In a comparison of two near-isoline maize (*Zea mays* L.) genotypes with differing osmotic accumulation, Beseli et al. [10] did find that the genotype with greater osmotic accumulation initially extracted soil water at a greater rate than the near-isoline without osmotic accumulation. As a consequence of the initially greater rate of soil water extraction, though, the near-isoline with osmotic accumulation reached the point of exhaustion of the soil water, which supported transpiration earlier than the near-isoline with low osmotic accumulation. The difference in the temporal pattern of soil water use meant that the isoline with greater osmotic accumulation had an earlier decrease in plant physiological activity, and this placed the plants in greater jeopardy from continued water deficit. Furthermore, the total water extracted from the soil was virtually unchanged, with plant osmotic accumulation due to the usual extremely small increases in extractable volumetric soil water content at low soil water potentials. The possible yield benefits of osmotic accumulation by plants in the field were experimentally shown to be directly dependent on the specific seasonal dynamics of water availability and its use, and the proposed benefit to increase yield as a result of osmotic accumulation was not clear cut. 

In addition, in many cropping circumstances, the maintenance of plant turgor by osmotic accumulation as water deficit develops is likely to yield results that are the opposite of what is beneficial to the crop. Turgor maintenance that allows continued plant development (particularly of the leaf area) and prolonged stomata opening ultimately makes plants more vulnerable as soil water content continues to decline at a high rate due to these characteristics. For many situations, the better response to a developing drought is for the plants to at least partially “shutdown” in order to decrease the rapid loss of water, conserve soil water, and allow sustained physiological activity over a longer period of water deficit. Simulations of soybean (*Glycine max* (Merr.) L.) production across the U.S. have shown the yield advantage of a crop that expressed a “partial shutdown” response to developing soil water deficit rather than osmotic accumulation, as the latter caused the plant to “ignore” the water deficit [11]. Not surprisingly, experimental studies of yield response to osmotic accumulation give quite mixed results [9,12], and this is due to the confounding interactions in the seasonal dynamics of soil water availability and plant water extraction. As Turner [9] pointed out: “despite the publication of >500 papers on osmotic adjustment since 1984 (www.plantstress.com, accessed on 27 September 2017) … there have been strikingly few reports of the release of new cultivars selected for high OA [osmotic adjustment] in any crop.”

### 5.3. Rate and Depth of Root Extension 

The reality of the response to altered root extension contains some of the same confounded response to cropping conditions as the ones that have been discussed for osmotic accumulation. Certainly, increased root extension into wet soil would be advantageous for drought conditions. However, plants with a greater capability for deeper rooting may prove non-useful under field conditions where chemical and physical barriers in the soil prevent greater depth of root extension. Once these soil barriers are reached, the acquisition of water by the crop is limited to the original water storage capacity in the soil above these barriers. 

Increased root proliferation also does not necessarily relate to more water extraction, since water uptake is increased a little once modest root length densities are reached. These root-length-density thresholds are fairly low, having been found experimentally in rice to be only 0.3 cm cm^−3^ in a uniform soil [13] and 2 cm cm^−3^ in the field [14]. Across species, Vadez [15] noted that the presence of more roots does not assure the adequate hydraulic characteristics that allow increased water uptake. 

A negative aspect of rapid root extension is that it may increase the risk of greater early-season depletion of soil water, which actually results in increased vulnerability to drought. In simulations of soybean growth across the U.S., Sinclair et al. [10] found a highly variable yield response by increasing the rate of soil penetration depth by roots from 20 to 25 mm d^−1^ (Figure 2, bottom right). For most locations, a greater root extension rate had an overall negative impact on yield, although in the driest regions, increased root extension had a positive result when the soil profile was initially water filled. 

### 5.4. Sustained Seed Set and Growth

Seed set and growth are often identified as particularly sensitive processes in grain yield production. Indeed, yields are commonly deceased substantially when drought occurs during reproductive development. However, the direct impact of water deficit on reproductive growth is better visualized by the ratio of the grain yield to total crop mass yield, i.e., harvest index. As it turns out, harvest index is fairly stable until the impact of drought on total crop growth is substantial. In maize, it has been found that there is a very close linear relationship between the decrease in seed mass and the decrease in total crop mass with water deficit [16,17] (Figure 2, upper left). The amount of decrease in seed mass with increasing water deficit was slightly greater than the amount of decrease in crop total mass. Hence, harvest index decreased in a curvilinear manner with increasing water deficit, and, as a result, major decreases in harvest index did not occur until drought was fairly severe. In the two cited studies with maize, there was not a large decrease in grain yield until the total crop mass decreased to less than about 1100 g m^−2^. Therefore, much of the impact of the initial loss in crop yield due to water deficit was essentially a result of an overall decrease in the ability of the crop to accumulate total crop mass. The critical opportunity to increase yield by sustaining seed development and growth exists only if severe drought has been shown to result in major decreases in harvest index. 

## 6. Challenges in Gene Modification

During recent decades, many investigations on plant response to water deficits and the most promoted results have been associated with the identification of single genes, or signalling pathways (mostly phytohormones), that confer “drought tolerance” [3,18]. While such results have been documented for specific controlled conditions, applicability in the field is a challenge. The difficulty here is that gene expression and the associated controlling networks for drought response are highly complex. Engineering of individual network components or of single genes to have durable, positive responses in overall plant-stress tolerance and yield stability has been elusive [19,20,21].

Molecular manipulations often overlook two essential issues: (1) scales of biological organisation, and (2) the temporal context of drought in the field. Crossing scales of biological organization by looking at the individual gene or, alternatively, by looking at at the pathway or signalling level and then the phenotype at the individual plant-organ level ignore all of the intermediate levels of biological organizations that lie between a single gene and a whole-plant expression (i.e., pathways, cells, and organs). Crucially, most studies ignore detailed phenological observations that allow resolution of the impact of any gene modification over a growing season at the crop level.

The second issue is a consideration of the stochastic and contextual nature of the development of soil water deficit in the specific climate and soil in which the crop is to be grown. There is no such thing as a unique, repetitious “drought” scenario for a location, but, rather, there is a suite of water deficit scenarios that vary among seasons in both intensity and timing. The crop response to these various drought scenarios can be quite variable depending on other weather variables, soil characteristics, and management practices. Hence, as stated by Tardieu [22], the “most relevant question on drought tolerance is probably not Does a given allele or trait confer drought tolerance” but, rather, “Does a given allele confer a positive effect on yield in an appreciable proportion of years/scenarios in a given area?” Answering this question requires a full understanding of the physiological consequences of gene transformation and, importantly, the yield response over the wide range of weather conditions over years that are experienced at any given location. Studies that impose rapid water deficit in pots under controlled environment conditions, which are often assessed by noting the irrelevant criterion of plant survival, do not offer the results necessary for extrapolating crop performance under realistic agronomic management in the field. 

Molecular studies tend to have a narrow temporal focus on genes controlling “stress-osmoprotectant metabolites”, such as proline and trehalose [23,24], or, alternatively, on fine-tuning stomatal conductance by manipulating the plant hormone abscisic acid (ABA), which is assumed to play a direct role in signal-transduction pathways and regulating stomatal closure [25,26]. Other studies have targeted the brassinosteroids pathway [27] that converges with the ABA pathway, or on auxins responses that modulate root architecture in order to boost water acquisition from the soil [28]. In all of these studies, the critical temporal dynamics of crop water use under the range of possible field scenarios for the development of water deficit are rarely considered.

Overall, the beneficial response of the molecular approach in field-grown plants is constrained for several reasons. (1) Many genes when upregulated turn out to have negative growth trade-offs [29]. (2) The target of these hormonal and signaling pathways lies in complex underlying molecular networks that have not been easy to decipher [30]. (3) Most of the studies on this were performed on non-crop species, such as Arabidopsis, under highly artificial water deficit conditions [30,31]. (4) The experimental setup of many of these studies assessed “drought tolerance” based on the irrelevant survival of severe water deficit [30].

## 7. Realistic Options for Increasing Effective Water Use under Water Deficits

There is complete consistency among the empirical analysis, the phenomenological description, and the mechanistic derivation regarding the notion that crop yield is ultimately constrained by the total amount of water that is available to the plants during the growing season. Such constrained yield possibilities are an important reminder that any plant modifications without a major increase in water availability can only be expected to result in small or, at best, moderate yield increases. That is, a realistic view of yield sensitivity to water availability indicates that only small absolute yield gains are likely in response to the “tweaking” of the physiology of the plant.

The physiological options for yield increase from a given amount of available water (Equation (2)) are illustrated in Figure 2. The size of the arrows in the figure connecting the mechanistic equation to the illustrations of each option indicate our estimates of the probability of grain yield increase under field conditions resulting from the alteration of the physiological trait. For example, as discussed previously, the possibility is low for altering the variable k in commercial cultivars to alter crop water use. Consequently, the probability of increasing yield by altering k is suggested to be low, as noted by the dotted-line arrow in Figure 2. That is to say, the biochemical composition of the grain and the photosynthetic pathways, barring the extremely challenging proposition of transforming a C3 species to C4 photosynthesis, offer a very limited opportunity for increasing the value of k.

It is suggested in Figure 2 that maintaining the harvest index at a high and stable value under water deficit offers a moderate opportunity for yield increase (upper left). Since major decreases in harvest index often occur only under severe drought conditions, attempts at stabilizing harvest index at high values seem likely to result in only very modest benefit under many water deficit conditions. Benefits from improved capability for set seed or growth can be expected only in the growing seasons when harvest index—not yield—of the current cultivars are consistently negatively impacted by drought. The dominating factor in yield decrease, even if the water deficit occurs during reproductive development, is still likely to be decreased overall crop growth due to inadequate water. 

The increase in water availability by altering rooting characteristics is suggested in Figure 2 (lower right) as another opportunity for a moderate increase in crop yield. However, in the field rooting options, this can be limited, as previously discussed. Water in the deep soil layers needs to be recharged by off-season rains so that water actually exists in the deeper soil layers and might be accessed by the roots. Of course, there must be no barrier in the soil that prevents the expression of deeper rooting. Finally, the seasonal dynamics of water use need to be regulated so that rapid root extension does not result in early-season use of water for greater vegetative development, which would cause the crop to be even more vulnerable to late-season drought.

Given that Equation (2) shows plant alterations of harvest index, k, and water recovery from the soil offer very modest opportunities for crop yield increase with water deficit, the remaining variable for the mechanistic expression of yield dependence on water use is VPD. While VPD is often viewed as an environmental variable beyond plant influence, the “effective” VPD under which water transpires by plants can be under plant control. This control results when partial stomata closure occurs at elevated atmospheric VPD (Figure 2, upper right, genotype Illusion). Genotypes that induce partial stomatal closure when VPD increases to the threshold atmospheric VPD for partial stomata closure will express soil water conservation. Early season soil water conservation becomes particularly important under conditions of late-season water deficit. The conserved soil water allows sustained physiological activity during the critical period of seed growth. The shift from lessened water use early in the growing season to conserve water for late-season crop growth increases the possibility of yield increase, as has been shown in simulations of soybean in the US [10] and Africa [32] and maize in the US [33]. Due to the temporal variability of atmospheric VPD on a daily and seasonal basis, it is suggested in Figure 2 that the sensitivity of stomata response to elevated VPD could have the largest impact on yield among the variables in Equation (2). 

At least a few genotypes in all of the major crop species have been identified that express the water conservation trait that results from partial stomata closure under elevated VPD [34]. As already indicated, simulations have predicted that the trait will generally increase crop yields. Another example of this is the simulation of wheat (*Triticum aestivum* L.) growth in Australia [35], where the results showed that the water conservation trait under elevated atmospheric VPD will lead to substantial reductions in post-anthesis drought stress and a higher average yield across the Australian wheat belt, especially in drought-prone environments (Figure 3). These calculations also showed that a positive Impact of this trait was expected to be especially beneficial under the conditions of climate change. The commercial AQUAmax line of maize hybrids, which express the water conservation trait, has higher yields in dryland areas, and it is being marketed by Pioneer Hi-Bred [36]. Soybean cultivars that express the trait have also been released for commercial production [37].

## 8. Symbiotic Nitrogen Fixation Sensitivity to Soil Water Deficits

Legumes have the unique capacity among crop species to fix atmospheric nitrogen into organic nitrogenous compounds in a symbiotic relationship with specific bacteria. While symbiotic nitrogen fixation is highly advantageous in crop production since little or no manufactured nitrogen fertilizer needs to be provided for the crop, high sensitivity of nitrogen fixation to soil water deficit can be a major issue [38]. For example, in soybean, a decrease in symbiotic nitrogen fixation rates commonly occurs at a high soil water content [39], which potentially results in a major limitation on crop yield formation.

The basis for the high sensitivity of some legume species to decreasing soil water content is not fully resolved, but the transport of nitrogen products from nodules seems to be an important factor. Nodules are a hydraulically closed system, insofar as water flux into the nodule is nearly all derived from the phloem flow into the nodule. The organic products of nitrogen fixation are then transported from the nodule using the water influx into the nodule in order to support xylem flow from the nodule. Any disruption of water flow either into nodules or out of nodules results in the accumulation of nitrogen products in the nodule, and this is associated with a feedback decrease in nitrogen fixation activity [38]. An important goal for improving legume productivity with drying soil is to identify genotypes with decreased nitrogen fixation sensitivity to soil drying.

Studies have been undertaken to identify genotypes in most major grain legumes that express nitrogen fixation resilience with soil drying. One of the largest studies was with soybean, in which a three-tier screening protocol was used [39] to identify genotypes that expressed nitrogen fixation insensitivity to soil drying. The initial screen was conducted on about 3000 well-watered genotypes in the field by selecting for a low-ureide concentration of petioles, which is an indication of nitrogen fixation drought resilience. About 10% of these genotypes were selected for direct screening of nitrogen accumulation in the field when subjected to 2 to 3 weeks of water deficit. Eighteen genotypes that expressed the highest sustained accumulation of nitrogen during water deficits were tested directly for nitrogen fixation capacity when subjected to controlled soil drying in a greenhouse. Eight genotypes were ultimately selected for their nitrogen fixation resilience to soil drying, and they have been used as parental lines in soybean breeding programs.

In addition to soybean, genotypic variability for nitrogen fixation tolerance to soil drying has been identified with grain legumes for common bean (*Phaseolus vulgaris* L.) [40], cowpea (*Vigna uniguiculata* L. Walp.) [41], and peanut (*Arachis hypogaea* L.) [42]. The identified genotypes with sustained nitrogen fixation activity at low soil water contents offer important genetic resources for breeding grain legumes in water deficit conditions. 

## 9. Concluding Remarks: Plant Improvement for Drought Amelioration

Major increases in yield require large increases in water input in the form of precipitation and/or irrigation. This is an important reminder that crop yield is quantitatively restricted by the amount of water available to the crop throughout the growing season. As it turns out, options for altered crop physiological characteristics that can improve the temporal dynamics of water use during a cropping season in order to improve yield are limited. As discussed above, several of the intuitive hypotheses sometimes suggested for improving crop performance under water deficits are likely to have little benefit or even a negative impact depending on the seasonal dynamics of water use in the field. One characteristic that has now been shown to generally result in yield benefit under many conditions, however, is soil water conservation as a result of partial stomata closure under elevated VPD. Water conservation early in the growing season, especially, results in greater water availability to the crop later in the season when a water deficit may develop. Increased water availability during seed growth, especially, has the potential to increase grain yield. In addition, increased nitrogen fixation resilience to soil drying in grain legumes has been shown to be an important trait for increasing grain legume yields.

## Figures and Tables

**Figure 1 plants-12-03137-f001:**
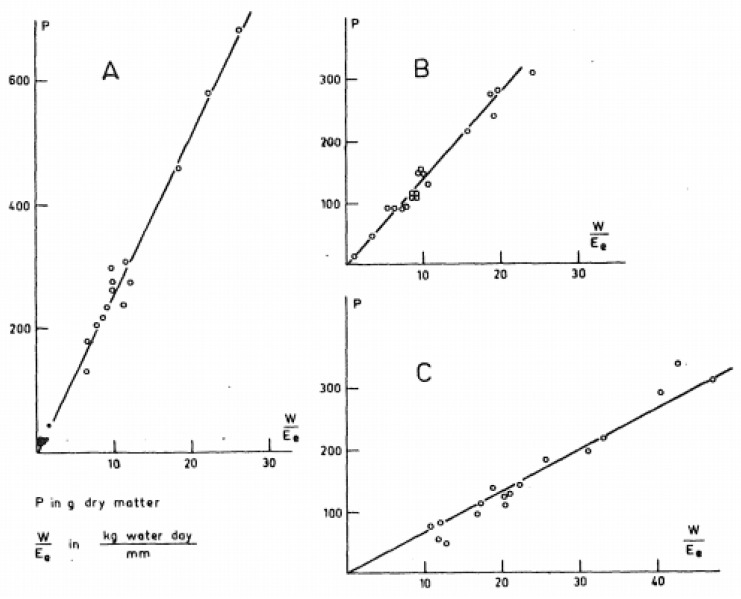
Graph developed by de Wit [6] of plant growth (P) versus plant water loss (W) normalized by water surface evaporation (E_o_) for (**A**) sorghum (*Sorghum bicolor* L.), (**B**) wheat (*Triticum aestivum *L.), and (**C**) alfalfa (*Medicago sativa *L.).

**Figure 2 plants-12-03137-f002:**
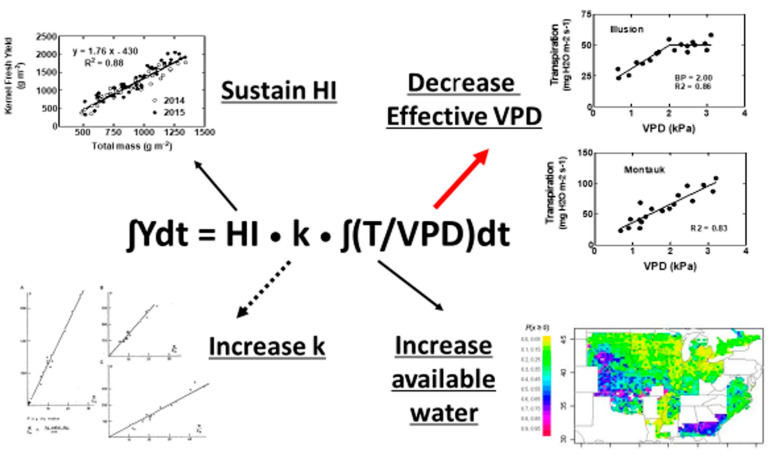
Figures of four realistic physiological options for increasing crop yield under seasonal water deficits. Among these options, the red arrow indicates the greatest probability for success in yield increase, the dotted arrow the least probability for success, and the black arrows an intermediate probability for success. (**Upper left**): Maize kernel yield vs. total crop mass showing a constant relationship over a wide of water and nitrogen treatments, taken from Jafarikouhini et al. [16]. (**Upper right**): Plant transpiration rate versus atmospheric vapor pressure deficit (VPD) for two cultivars of maize differing in response at elevated VPD from Beseli et al. [9]. (**Lower left**): Constant slope within species found by de Wit [6] reflecting constant k for a species. (**Lower right**): Map of probability of soybean yield simulated for increasing root extension rate from 20 to 25 mm d^−1^ from Sinclair et al. [10]. Only the areas in the map shown in blue and red were found to have a probability of yield gain greater than 0.50.

**Figure 3 plants-12-03137-f003:**
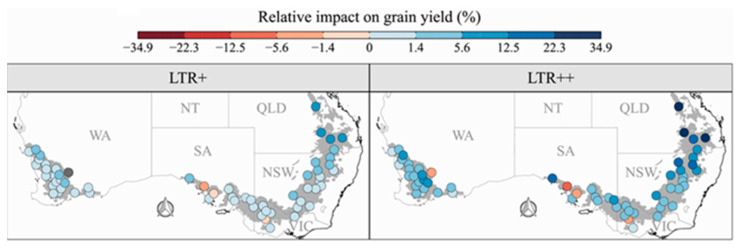
Collins et al. [33] projected average yield increase for 60 locations across the Australian wheat belt for which plant transpiration rate was decreased when VPD exceeded 1.3 kPa. Two genotypes were simulated, and the transpiration at elevated VPD decreased by 30% (LTR+) or 100% (LTR++).

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
