# Peer review of "Realistic Physiological Options to Increase Grain Legume Yield under Drought"

_plants, 2023, doi:10.3390/plants12173137_

Round 1
Reviewer 1 Report
This work treats the options we have to increase the yield of legume crops on drought conditions.
The document is well written, the introduction provides good reasons to read the rest of the document, even if some more references should be included in lines 29-32; 32-33; 35-36 and 39-40
The main development of the idea based in the physiological equations is original and clear, the conclusion are supported by the previous developments and justified by the litterature results.
Then minor corrections should be perfomed:
Line 12: reconsider rewritten or maybe changing however by often
Lines 90 & 91, it is not clear what you mean by adding "Pa"
Line 144 check, probably you need to supress "the" before "a"
Figure 2: you talk in the legend about the "size" of the arrows, but in the figure it's not really clear as you used also a red arrow and a dotted arrow.. please review the form to make it more clear
Line 254: change gran by grain
The form of the bibliographic references in the text must be changed to numbers as in the list.
Author Response
We appreciate the favorable comments made by this reviewer concerning our manuscript. The reviewer offered a few minor suggestions for revision and these were all incorporated into the revised manuscript.
Reviewer 2 Report
Dear authors,
In attach is the PDF with some suggestion.

The manuscript needs some revision.
Author Response
We appreciate the suggestions made by this reviewer. The reviewer identified several sentences to be restructured to better describe the ideas being presented. In each case, the sentences were restructured in the revised manuscript and we hope the restructured sentences offer greater clarity for the ideas being presented.
Round 2
Reviewer 2 Report
Dear authors,
Unfortunately, you did not perform several of my previous comments. For my mistake, I attach the wrong file in last revision however I sent to the editor. Probably you did not understand. So, I send in attach the previous pdf file and the new reviewed file. Please, see both.
Other important point, the manuscript did not refer the objetive and the title is not reflect of the manuscript.
After big and careful revision, I think the manuscript can be accepted to publish.

Round 3
Reviewer 2 Report
I read the manuscript, and the authors need to be careful with some formatting errors.After the correction of these mistakes and if the Academic Editor agree the manuscript can be accepted to publication. I read the manuscript, and the authors need to be careful with some formatting errors.
After the correction of these mistakes and if the Academic Editor agree the manuscript can be accepted to publication.
Author Response
Format revisions were made in manuscript.
